# Patient preferences in genetic newborn screening for rare diseases: study protocol

Sylvia MARTIN ![ORCID] ,[1] Emanuele Angolini,[2] Jennifer Audi,[3] Enrico Bertini,[2] Lucia Pia Bruno,[4,5] Joshua Coulter,[6] Alessandra Ferlini,[7] Fernanda Fortunato,[7] Vera Frankova,[8] Nicolas Garnier,[6] Åsa Grauman,[1] Edith Gross,[9] Brett Hauber,[6] Mats Hansson,[1] Janbernd Kirschner ![ORCID] ,[10] Ferdinand Knieling,[11] Gergana Kyosovksa,[12] Silvia Ottombrino,[2] Antonio Novelli,[2] Roman Raming,[11] Stefaan Sansen,[13] Christina Saier,[14] Jorien Veldwijk[15]

**Correspondence to**
Dr Sylvia MARTIN;
sylvia.martin@crb.uu.se

## ABSTRACT

**Introduction** Rare diseases (RDs) collectively impact over 30 million people in Europe. Most individual conditions have a low prevalence which has resulted in a lack of research and expertise in this field, especially regarding genetic newborn screening (gNBS). There is increasing recognition of the importance of incorporating patients' needs and general public perspectives into the shared decision-making process regarding gNBS. This study is part of the Innovative Medicine Initiative project Screen4Care which aims at shortening the diagnostic journey for RDs by accelerating diagnosis for patients living with RDs through gNBS and the use of digital technologies, such as artificial intelligence and machine learning. Our objective will be to assess expecting parent's perspectives, attitudes and preferences regarding gNBS for RDs in Italy and Germany.

**Methods and analysis** A mixed method approach will assess perspectives, attitudes and preferences of (1) expecting parents seeking genetic consultation and (2) 'healthy' expecting parents from the general population in two countries (Germany and Italy). Focus groups and interviews using the nominal group technique and ranking exercises will be performed (qualitative phase). The results will inform the treatment of attributes to be assessed via a survey and a discrete choice experiment (DCE). The total recruitment sample will be 2084 participants (approximatively 1000 participants in each country for the online survey). A combination of thematic qualitative and logit-based quantitative approaches will be used to analyse the results of the study.

**Ethics and dissemination** This study has been approved by the Erlangen University Ethics Committee (22–246_1-B), the Freiburg University Ethics Committee (23–1005 S1-AV) and clinical centres in Italy (University of FerraraCE: 357/2023/Oss/AOUFe and Hospedale Bambino Gesu: No.2997 of 2 November 2023, Prot. No. _902) and approved for data storage and handling at the Uppsala University (2022-05806-01). The dissemination of the results will be ensured via scientific journal publication (open access).

## STRENGTHS AND LIMITATIONS OF THIS STUDY

⇒ This study has been developed by an international expert team together with industry and patient representatives to provide an example of a rigorously designed two-stepped preference study.

⇒ This study will collect data from a significantly large sample across several populations, allowing comparison of genetic newborn screening (gNBS) uptake perspectives, attitudes and preferences, psychological variables and socio-demographic information in two European countries.

⇒ Only the most important characteristics of gNBS can be incorporated into the discrete choice experiment (DCE); therefore, considerations of less important characteristics can only be interpreted based on the outcomes of the qualitative work and have to be assumed less important (on average) than the least important outcome of the DCE.

⇒ Study elements might be complex (cross-cultural elements, potential issues in translations, recruitment population differences) and prone to misunderstanding even if high involvement of patient representatives in the development of the study material will reduce these risks.

⇒ This study will recruit participants from different countries where the in-place gNBS procedure is differently implemented in general care settings.

## INTRODUCTION

There are more than 6000 known rare diseases (RDs), conditions that affect one, or less than one, individual in 2000 (point prevalence of $<1/1\,000\,000$ in more than 84% of RDs[1]). People living with rare a disease (PLWRD[i]) typically face an arduous journey to proper

---

[i]This acronym has been selected based on its usage in international literature https://www.rarediseasesinternational.org/wp-content/uploads/2022/01/Final-UN-Text-UN-Resolution-on-Persons-Living-with-a-Rare-Disease-and-their-Families.pdf

diagnosis, enduring on average 8 years of countless physician consultations.[2] Time to diagnosis is a stressful period during which patients are faced with inconclusive test outcomes, possible misdiagnoses and thereby ineffective treatments and healthcare resource utilisation.[3]

Although RDs collectively impact over 30 million people in Europe, most individual conditions have a low prevalence.[4] This has resulted in a lack of research and expertise in this field. Difficulties in the process of receiving an accurate diagnosis and uncertainty around experienced symptoms continue to add to the difficulties that PLWRD experience, but also affecting their families, caregivers, physicians and society as a whole.[5 6] Currently, there is increased recognition of the importance of incorporating patients' needs and perspectives into the shared decision-making process and providing more avenues for engagement and involvement of PLWRD at each stage of the design and development of health interventions. Consequently, PLWRDs play a vital role in the implementation of novel solutions that are designed to improve their own life and health.

The rapid evolution of genetic diagnostic techniques, including gene panels and whole-exome and whole-genome sequencing, will provide new opportunities for early diagnosis of RDs potentially enabling PLWRDs to also have better access to therapies and treatments. Thus, genetic newborn screening (gNBS)[ii] could lead to a public health paradigm shift with early diagnosis and intervention which may prevent health damage before it is irreversible and avoid functional limitations leading to premature death in potentially preventable diseases. Implementation of gNBS for RDs is thought to improve early access to care, redirect/focus treatment,[7 8] impact pregnancy and family planning[9] and increase the detection rate and accuracy of newborn screening programmes as a whole.[10]

However, undergoing gNBS has obvious ethical and social implications both for newborns and for parents as it may induce stress and increase concerns regarding diagnosis, treatment and the child's future.[11 12] Ethical and psychosocial issues related to gNBS present considerable dilemmas regarding the effects of genetic data on both the infant and their family, as well as with regard to broader societal aspects. These concerns play a critical role in shaping regulatory structures and health policies aimed at addressing these ethical obstacles (Reinsteing, 2015; Grob, 2019).[13 14] The National Human Genome Research Institute *Newborn Screening Fact Sheet* states, 'with the decreasing cost of genome sequencing, there is potential for its clinical application in newborn screening. This could supplement or replace traditional panels of tests, providing more comprehensive health information. However, several questions remain about the practicality, ethics, and long-term implications of incorporating genome sequencing into routine newborn screening' (https://www.genome.gov/about-genomics/fact-sheets/Newborn-Screening-Fact-Sheet, consulted on 2 January 2024). Therefore, clarifying healthcare professionals' (HCPs), patients' and general public perspectives, attitudes and preferences on gNBS implications may improve its safe and equitable development.

Additionally, there are challenges related to the interpretation of gNBS results, especially if based on whole-exome or whole-genome sequencing approaches: communication of test results, management of incidental findings and/or of variants of uncertain significance, data storage and other issues that need to be addressed.[15 16]

Bearing in mind that gNBS and its results may impact society and the individual itself, their adoption and implementation in medical practice depends on attitudes, experiences and preferences of different stakeholders. This study aims to understand the general public's perspectives and attitude on gNBS and their preferences towards gNBS. Of particular interest are any differences in perspectives, attitudes and preferences between medical geneticists, expecting parents seeking genetic consultation during their pregnancy and 'healthy' expecting parents.

There has been an increase in preference studies on genetic screening. These studies examine various aspects and challenges from both patients and HCP stakeholders.[17] Even if the general public's attitude towards gNBS for RDs tends to be positive,[18–20] research also demonstrated that the acceptance of gNBS in the general public does not reflect the uptake of gNBS during pregnancy.[21]

The first part of our protocol describes a systematic literature review of preference study results for gNBS (PROSPERO: CRD42022297678) in order to inform the design of the qualitative and quantitative parts of the protocol. Specific methods like discrete choice experiment (DCE) for eliciting preferences for gNBS have not been thoroughly implemented in Europe, and only two previous DCE studies in this field were identified. Miller *et al* assessed the perception of lay audiences regarding gNBS in Canada.[22] Respondents from the general public were positive about the potential for gNBS expansion in the country based on the expected clinical benefits, improvement of reproductive risk management and also the possibility of earlier diagnosis. Wright *et al*, in another DCE, investigated information provision among midwives in the UK, when asking them about when to disclose information to expecting parents or what type of information should be provided.[23] According to this study, the potential for receiving a 'false positive' result should not be disclosed (as it may not be helping decision-making), and the best period to provide information would be late pregnancy to 3 days post-birth. The current study protocol, 7 years after the first DCEs on the

---

[ii]We will use a broad definition of genetic newborn screening as defined by the Institute NHGR. 2023, 17th November [Available from: https://www.genome.gov/genetics-glossary/Newborn-Genetic-Screening : "*Newborn screening is a set of laboratory tests performed on newborn babies to detect a set of known genetic diseases. Typically, this testing is performed on a blood sample obtained from a heel prick when the baby is two or 3 days old. [...].*"

topic in Europe, offers a comparison point for diverse cultural perspectives, attribute importance and technical advances in genetic technique preferences.[22] Moreover, this study will allow us to assess the perspective of gNBS in Europe among expecting parents who might face such a decision as healthcare systems are on the verge of implementing or extending gNBS programmes. In fact, most of the European countries that implemented neonatal screening in the 1960s/1970s are now implementing an extended panel, envisioning the potential to screen for 40–50 conditions to be tested with a single blood spot using improvements in molecular technologies.[24] Protocols for gNBS research offer innovations for prompt and efficient treatment, as well as reducing obstacles related to worry in parents or overdiagnosis (Gray et al., 2008).[25] The research about gNBS serves as a catalyst for ongoing advancements in the screening's sensitivity and specificity. The goal of the screening process optimisation efforts is to provide assurance that true positive cases will be accurately diagnosed and to prevent families from being upset by an unexpected false positive result or from experiencing severe anxiety as an effect of results with uncertain significance (La Marca et al., 2023).[26] This study will inform both clinical research and patient advocate stakeholders contributing to gNBS for RD field.

A recent international research protocol (six countries) addressed the preferences of women towards prenatal screening results,[27] with the same stepwise structure as our protocol but in a connected field (prenatal testing). Based on a systematic literature review, they extracted 19 tentative attributes and refined them into 12. On the same prenatal topic, Buchanan et al evaluated women's decisionmaking on prenatal genomic screening with a DCE (12 scenarios).[28] These results were collected on an international (eight countries) level including a wide range of cultural examples from China to the United States and showed that women looked for high diagnostic yield, short turnaround times and uncertain results to be reported (both variant of unknown significance and secondary findings).

These attributes and choices may be considered for neonatal gNBS, but they may be different as the pregnancy planning decisions are not at stake anymore, and the attributes of importance for new parents may be different after the birth of their infant. Our research will assess the perspectives, attitudes and preferences of a more targeted population of expecting parents.

## METHODS AND ANALYSIS
### Research aims and objectives
This study is part of the larger project entitled 'Screen-4Care (S4C)'. S4C is a research project launched in October 2021, which aims at shortening the diagnostic journey for RDs by accelerating diagnosis for PLWRDs through gNBS and digital technologies such as artificial intelligence and machine learning. S4C has received funding from the Innovative Medicines Initiative 2

(IMI2) Joint Undertaking (JU). IMI, now superseded by Innovative Health Initiatives, is the world's largest public-private partnership in the life sciences whose mission is to 'improve health by speeding up the development of, and patient access to, innovative medicines, particularly in areas where there is an unmet medical or social need'.

The expected start of the study is December 2023 ith a planned end in July 2024. The study has several objectives:
- ► To measure perspectives and attitudes of medical geneticists, parents seeking genetic consultation and 'healthy' expecting parents for gNBS
- ► To elicit preferences of 'healthy' expecting parents from the general population as well as of parents seeking genetic consultation
- ► To identify and understand any heterogeneity in preferences for gNBS based on demographic characteristics and attitudes towards screening and genetic testing in general
- ► To provide predicted uptake rates of gNBS based on the willingness to take part in RD screening programme based on predefined characteristics of the screening test and output (not disease specific)

### Patient and public involvement
Research questions and information materials have been developed with patient representatives.

### Design of the study
A DCE technique will be used in this study, which is a globally established technique to elicit preferences in areas such as marketing, environmental economics, transportation and health economics.[29–32] DCEs are based on Random Utility Theory.[33–35] Respondents are asked to complete several 'choice tasks' (online), each of which consists of two or more alternatives that describe a treatment at hand. This description is based on treatment characteristics (ie, attributes). It is assumed that an individual's preference for an alternative is based on the values of the attribute levels. Individuals' preferences can be inferred from the choices made across multiple choice tasks. Using a DCE, researchers can quantify the importance of preferred treatment characteristics, calculate willingness-to-pay or willingness-to-accept risks and estimate potential participation rates, which makes this method highly attractive and superior to competing methodologies.[29 30] An example of a choice task for our study is included in table 1. The design and analysis of this survey will follow best practices.[36 37] Since a DCE can only include a total of about 5–7 of the most important attributes, it will not provide a fully holistic overview of the importance of all characteristics of gNBS on the decision to participate in such a screening procedure. However, using only the most important attribute results, uptake behaviour can be correctly predicted as the positive predictive value of DCEs has been shown to be over 0.90 (De Bekker-Grob et al., 2020)[38]

This prospective, non-interventional, cross-sectional study will be designed in a stepwise manner.[32] First, a

**Table 1** Example choice task DCE

| | Screening A | Screening B | No screening |
|---|---|---|---|
| Accuracy of screening test | 90% | 85% | Not applicable |
| Waiting time between test and results | 2 weeks | 1 month | NA |
| Healthcare professional informing you about results | Nurse | General practitioner | NA |
| I prefer: | 0 | 0 | 0 |

Note: This figure is for illustrative purposes only. The final choice task will be determined after considering information learnt from the interview and focus groups. A 'no screening' alternative will be included in the DCE study since this option is necessary to estimate the uptake of screening alternatives, including the proportion who would opt out of screening.

systematic literature review will be performed to develop an initial assessment of the perspectives and attitudes of HCPs and parents regarding gNBS for RDs. Data extraction forms were designed by researchers involved in this process (11 duos of experts from the S4C project). Following the initial selection, an external firm specialising in various databases and JBI tools, LUCID Ltd, located in Marlow, England, conducted a scientific quality evaluation (appraisal scoring). Subsequently, only those articles that exhibited a quality level above the average in their respective category were selected for further analysis. Data were analysed using thematic analysis. The main outcome of the review was a list of characteristics of gNBS that impact decisions of parents to participate in such screening or not (ie, the list of potential attributes)[iii].

The qualitative research step will gather data using both focus groups and individual interviews. These two different types of interviews will collect both explorative and in-depth data. Focus groups will be conducted with expecting parents seeking genetic consultation as well as 'healthy' expecting parents, while individual interviews will be conducted with medical geneticists. The individual interviews and focus groups are expected to start in 2023 and to be completed before June 2024. During all interviews, a structured interview guide will be followed by trained interviewers. Before the start of the interviews, participants will be asked to sign an informed consent form (paper version if the group is held in person or online form version if the group is held via video conference) and complete a short demographic survey (age, gender, number of previous pregnancies (expecting parents only) and years of experience (clinicians only)). The focus groups and individual interviews will follow the structure of the nominal group technique (see[30]). After several introductory questions, participants will be asked to rank a list of potential attributes (emanating from the literature review) from most to least important according to their personal willingness to take part in gNBS. These rankings will be discussed, and following the discussions, participants will be allowed to change their ranking.

Patient representatives from 11 organisations, who are a member of the S4C Patients Advisory Board (PAB) and joined the consortium to ensure that the perspective of PLWRDs is secured and given priority in all research activities of S4C, will review the interview guides and short demographic survey to provide feedback and make sure the wording and definitions used are understandable and appropriate.

Based on the attribute ranking in the qualitative portion of the study, the DCE will be designed. This DCE will be part of a larger survey including background questions and clinical scales (see 'Data analysis' section). Given the number of attributes and levels included in the DCE, respondents would not be able to complete a full factorial design (ie, over 1000 choice tasks). Most DCEs have respondents answer 8–15 choice questions. Blocked designs offer the opportunity to increase the efficiency of the data collection, while the cognitive load on respondents stays low as they only complete a subsection (block) of the total number of choice tasks. Therefore, such designs are typical in DCEs.[31] When blocking a design, all respondents will still see all attributes and levels but only a subset of combinations. The blocks of questions each respondent receives will be randomised to mitigate any potential bias of the blocked design while maximising statistical power of the full design in the total study population.[29]

The DCE itself will be constructed using a Bayesian D-efficient design with choice tasks consisting of two hypothetic gNBS alternatives and a 'no screening' alternative.[29 39] The educational section of the survey will be designed in close collaboration with patient organisation partners (PAB and EURORDIS representatives) similar to the interview guides, to ensure that the information is accurate and clear to all participants and ensure readability and appropriateness of terms. Following the development of the instrument, the survey will be pretested among expecting parents seeking genetic consultation, 'healthy' expecting parents and medical geneticists to ensure that the survey is understandable to the same population that will take the final online programmed survey in domestic languages.

### Study population and recruitment

For the qualitative part of this study, three groups of interviewees are identified: (1) medical geneticists (individual interviews), (2) 'healthy' expecting parents from the general population (focus groups) and (3) expecting parents

---

[iii]The output of the systematic literature review is planned to be published separately as a stand-alone paper in a scientific journal.

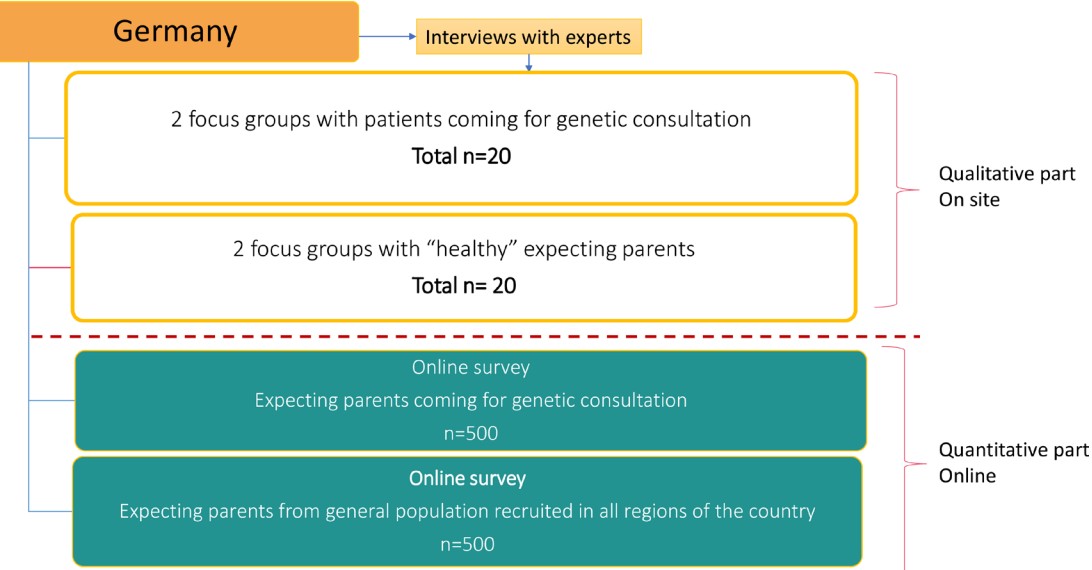

**Figure 1** Study flowchart in Germany.

currently seeking genetic consultation (focus groups). These three groups will be recruited both in Germany and in Italy. All participants must be between the age of 18 years and 70 years. Furthermore, they must be able to understand and speak German or Italian as all interviews and survey material will be provided in native country-specific language (German or Italian). German and Italian procedures and methods are illustrated in figures 1 and 2.

### Distribution of participant group to study phases

Medical geneticists are considered to be experts in precise genetic screening, advanced interpretation and result delivery to HCP teams and patients. Medical geneticists involved in genetic screening will be recruited via purposive sampling within clinical partners of S4C project and be interviewed as experts. The results of these interviews will not lead to any specific analyses but ensure the adequacy of the interview guide and questions.

The second group consists of 'healthy' expecting parents from the general population (with pregnancy monitoring in regular care) recruited in the same geographic areas as the expecting parents currently seeking genetic consultation for RDs in one of the clinical centres. The focus group will be run by trained personel at the recruiting centre (trained midwives or research nurses, depending on available resources). Uppsala University will support the implementation and conduct of the group interviews. As a precautionary measure, if both partners volunteer in the study, the mother and father from the parental couple will be included in separate groups to avoid any influence they may have on each other during the discussions.

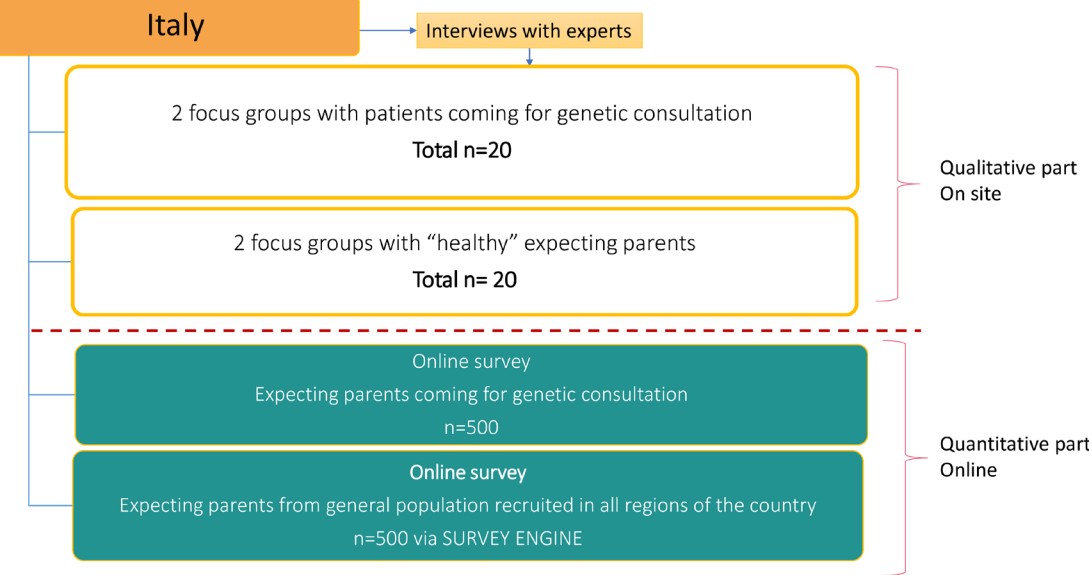

**Figure 2** Study flowchart in Italy.

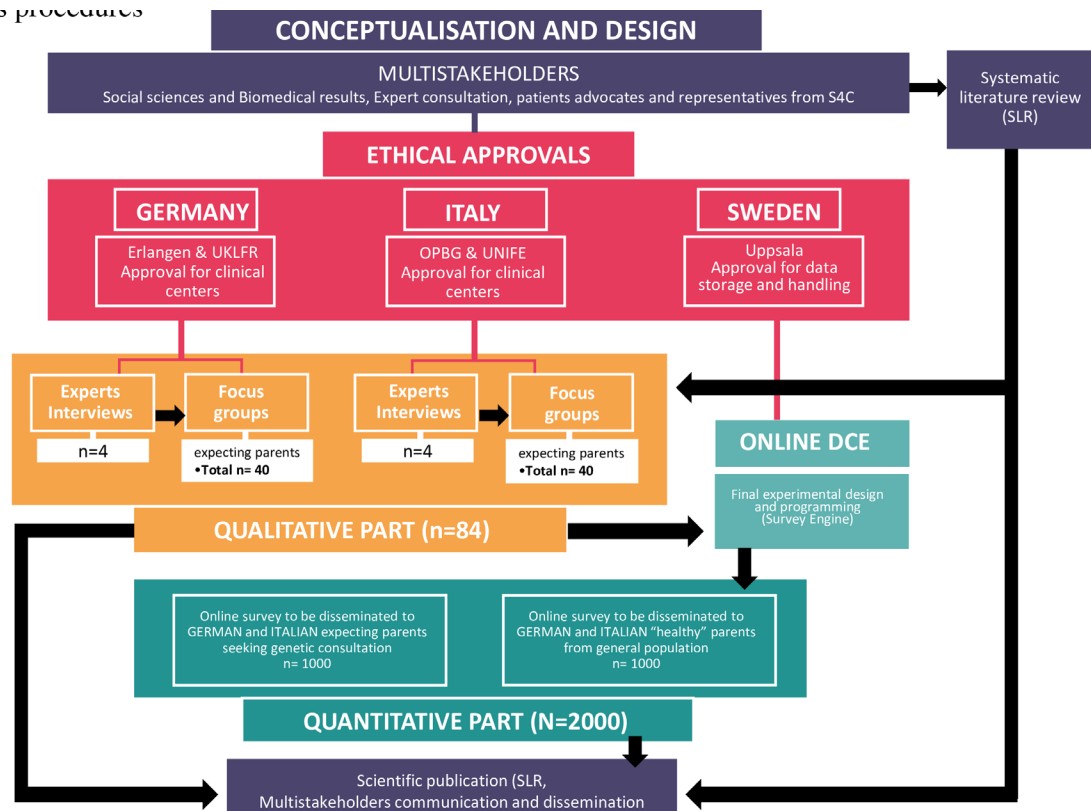

**Figure 3** Protocol's procedures.

The recruitment of 'healthy' expecting parents will be conducted via midwifery and gynaecology practices in Germany and in Italy.

The third group of participants will include expecting parents who seek genetic consultation but who are not necessarily carriers of genetic variants related to RDs themselves. The recruitment of expecting parents seeking genetic consultation will be conducted by clinical genetic centres that are part of the S4C project in both Germany and Italy. Parents will be identified as they visit the clinical centres.

For the DCE survey, 'healthy' expecting parents from the general population and expecting parents already seeking genetic consultation will be recruited. 'Healthy' parents from the general population will be recruited via SurveyEngine[iv] in Italy and via invitation shared from participating centre networks in Germany. Parents already seeking genetic consultation will be recruited via invitation from the participating clinical centres in Germany and Italy. The study aims to recruit approximately 2084 participants in total across the two participating countries (see figure 3). For the qualitative phase, a maximum sample of 4 experts (2 in Germany and 2 in Italy) and 40 expecting parents (representing 20 couples in each country) for the group interviews will be recruited. The recruitment will be operated by the clinical centres that

have access to 'healthy' expecting parents (a total of 84 for the first part). For the quantitative phase, the survey sample will be about 1000 participants in each country (500 'healthy' expecting parents from the general population and 500 expecting parents seeking genetic consultation) (see figures 1 and 2).

A priori sample-size calculations represent a challenge in DCE experiments. Most published choice experiments have a sample size of 10 to more than 3000 respondents.[40] However, minimum sample size depends on several criteria, including the question format, the complexity of the choice task, the desired precision of the results and the need to conduct subgroup analyses.[36] A method for computing sample size was proposed by de Bekker-Grob et al[37]; however, as the article points out, there is no analytic solution or power calculation that can be used to determine the appropriate sample size for a DCE unless the researcher has enough information to inform the selection of priors. Additionally, for the 'healthy' expecting parents from the general population sample, we plan to use a blocked design (asking people to only complete a randomly assigned subset of the choice tasks, thereby limiting the cognitive burden on each participant). Given what we know, recruiting 1000 respondents in each country should be sufficient to answer the proposed research questions and provide enough information to identify preferences in these groups and comparisons across these groups with acceptable precision.

---

[iv]SurveyEngine is a third-party recruitment company. https://survey-engine.com/

## Adapting methods to ethical specificities

German and Italian regulations differ in their understanding of the potential risk coming from the participation to a focus group in terms of privacy and discomfort for the participants (eg, genetic-related conversation that could lead to sharing opinions or experiences connected with genetics in front of other participants might be an argument for ethical committees' refusal in Italy). Methods have been adapted, for example, by conducting individual interviews for parents seeking genetic screening (more sensitive sample) only by a clinical geneticist (in Italy). Any informational material provided before signing the informed consent will point out the specific risks and expected benefits for participants to make an informed decision before entering the protocol.

## Study procedures

### Information of participants and informed consent

Each participant will receive detailed written information on the nature, objectives and possible risks/benefits of the research (for both the focus groups and quantitative preference study participation). Prior to the qualitative interviews, participants will be given time to consider their involvement and ask questions about the study to the researchers. They will be informed that they can stop their participation in the study at each point. Participants can only take part in the study if the informed consent form is signed beforehand. The investigators will keep the original signed informed consent in accordance with current requirements. For online survey participation, the informed consent will be included in the online survey and digitally signed by the participants.

### Individual interviews

Semistructured interviews will be conducted with members of the IMI S4C consortium to test the interview guide and associated questions. These interviews will be conducted in English. The interview will take approximately 1 hour and will be conducted at a location convenient for the interviewee, via telephone or via video conferencing platform.

Feedback from the pilot interviews will be evaluated, and the interview guides will be adapted accordingly for each country. After making the necessary changes, the interview guide will be translated into German and Italian. Native language-speaking trained interviewers from clinical centres will conduct the interviews using the interview guide. Collection, recording and reporting of data will ensure the privacy, health and welfare of research subjects during and after the study and in accordance with the country-specific regulations.

### Group interviews

The group interviews will be used to provide more precise evidence for the most important characteristics of gNBS when parents decide about their participation in genetic screening programmes.

At the start of the interview, the objective of the S4C project and the interview procedures will be explained to the interviewee, and the researcher will confirm that informed consent was given before starting the interview. The interviewee must sign the consent form before the start of the interview. Once informed consent is confirmed, the researcher will start the recording device. After asking the questions in the interview guide, participants will be asked to rank a list of potential attributes (emanating from the literature review) from most to least important according to their personal decision to take part in gNBS. These rankings will be discussed, and following the discussions, participants will be allowed to change their ranking. The interviewer will ask the interviewees if they want to add something to the interview.

We plan to conduct a minimum of four group interviews per country (Germany and Italy) (see figures 1 and 2, and for overall description, see figure 3) and continue until data saturation. There will be eight group interviews, in total. A group interview consists of 8–10 members. The group interview will take about 2 hours and will be conducted at a location and on a date most convenient for all participants.

The interviews will be recorded and transcribed verbatim. The interviews will be analysed iteratively using a grounded theory approach based on a mix of inductive and deductive coding. Analyses will be conducted in *Atlas.ti* software. The coding procedure will be handled by two independent coders from each country. The outputs will then be translated into English for further use by the research team. To ensure a proper standardisation of the interviews across languages, the domestic researchers will be trained by Uppsala's university researchers and will also be provided a detailed interview guide to help them through the process of every interview.

The information given by the interviewee will be treated as confidential and will be processed anonymously. The collected data will be stored securely and viewed only by authorised researchers of the project. The processed answers will be used for publication of reports or articles as part of the S4C research project. The audio recordings specifically will be destroyed after completion of the study, and no files will be stored at the recruiting centres.

### Discrete choice experiment

A survey will be fully developed after the final selection of attributes and levels. The final selection of attributes (and levels) to be included in the DCE will be based on the (focus group) interviews with medical geneticists and PLWRD's advocates. First, attributes ranked highest during the qualitative studies will be taken forward for consideration. For attributes to be included in the final DCE, they should be unambiguous, not overlapping and clearly defined. Additionally, they should be important to the anticipated target population while at the same time clinically relevant. Therefore, the final selection of the most important attributes will be based on attribute ranking from the focus groups in combination with the

perspective of experts on practicalities of gNBS. After selecting the final attributes, appropriate levels that reflect current practice will inform the attribute list for the DCE.

The survey instrument, including the DCE, will be pretested using think-aloud methods in a convenience sample of 3–5 'healthy' expecting parents from the general population in each country, using think-aloud interviews to review the results from the interviews and to determine if any adjustments need to be made to improve clarity. This process will be conducted in collaboration with EURORDIS and patient organisation representatives of the PAB. They will ensure accurate use of vocabulary by assisting in drafting and reviewing the included definitions and explanatory texts as well as educational content to ensure accessibility and readability to lay audiences.

The initial DCE experimental design will be generated via NGene 1.0 (ChoiceMetrics, 2011), and hypothetical choice questions with three alternatives will be created (of which one is the 'no screening' alternative). These alternatives will be characterised by varying attribute levels. To ensure design efficiency, a pilot with 100 respondents in each country will be conducted. All participants will provide informed consent before entering the task. Based on the data retrieved in the pilot test, a multinomial logit model will be fitted. Beta estimates will be used to assign priors for the final experimental DCE design, which is a *d*-efficient (Bayesian) design.[33] To increase the efficiency of the design, multiple blocks of choice tasks will be generated which will later be randomly distributed to respondents in such a way that each respondent will answer only one block of approximately 10–15 choice tasks.

### Ethical and legal aspects
The S4C IMI Project (grant agreement no. 101007757) has been approved by the Medical Ethics Committee of References.

### Ethics and dissemination
This study has been approved by the Erlangen University Ethics Committee (22–246_1-B), the Freiburg University Ethics Committee (23–1005 S1-AV) and clinical centres in Italy (University of FerraraCE: 357/2023/Oss/AOUFe and Hospedale Bambino Gesu: No.2997 of 2 November 2023, Prot. No. _902) and approved for data storage and handling at Uppsala University (2022-05806-01). The dissemination of the results will be ensured via scientific journal publication (open access).

The ethical procedure will be performed in two distinct stages, the first for the qualitative part and the second for the quantitative part (DCE). As the first part will be used as an input and will inform the second one, ethical committee may not consider approving a 'tentative' survey draft or a 'tentative' list of attributes as it can significantly change the ethical risk for anonymity (if one attribute is 'cost' compared with 'pregnancy planning'). The overall S4C project also ensured an ethical framework with the Code of Ethics Practice that was developed internally as a project deliverable and agreed with all Work Package leaders to inform each researcher within the S4C consortium about the main ethical risks with no exception of this specific task on preference studies.

### Confidentiality
Due to the interaction of the researchers with the interviewees, it is impossible to collect the data anonymously from the recruiting centres, but the obtained information will be coded to allow confidential and anonymous processing and reporting of the data from the research team. The identity of the participants will not be revealed, so privacy will be guaranteed. The research data will be used for publications in scientific journals and writing of reports or publication for the S4C project but will be registered and saved with a secure code. The coded anonymous data can be shared with the members of the consortium. The code, being the identification numbers linked to the identity of the participants, will be held confidential within the group of involved researchers. Only the researchers who are involved in this study will have access to this code, and only they will know the identity of the participating stakeholder members.

Uppsala University will provide secure data storage space with restricted access to a researcher analysing the data from the S4C partners involved in task 3.1. All digital data will be stored in Uppsala University's official storage of research data (data portal Allvis) with specific ethical approval from the Swedish Ethical Review Authority, as Swedish Regulation requires. The data are physically stored in two redundant data centres following (non-certified) ISO 27001 guidelines. Both data centres are equipped with an enterprise-grade network, servers and hard disks with high availability, security and safety. All data centres are equipped with access controls, cooling systems, backup power and surge protection. They are protected against common threats such as fire, water damage, burglary and theft. Access is restricted to technical staff and strictly protocolled. All data are encrypted in transit (https) and at rest (through database encryption). Allvis is configured with versioning of files, saving up to 500 versions of an individual file. A recycle bin saves deleted files for up to 30 days. Content databases are configured with transactional logging, limiting potential data losses to 1 hour during critical failures. Databases are backed up incrementally, daily, weekly, monthly and yearly using a backup system hosted in the same data centres. Network traffic is protected through a robust firewall, and servers hosting file content are not directly connected to the internet. The involved researchers commit to the highest standards of data security and protection to preserve the personal rights and interests of study participants. Participants will be informed that they can contact Uppsala University data security services for any information via email (dataskyddsombud@uu.se).

The involved researchers commit to the highest standards of data security and protection to preserve the personal rights and interests of study participants. They will adhere to the provisions set out in:

(1) The General Conference of UNESCO, meeting in Paris, from 9 to 24 November 2021, at its 41st session for Open science recommendations.

(2) A central guide for all research that includes human subjects will be the Declaration of Helsinki ethical principles for medical research involving human subjects from the World Medical Association (WMA, adopted by the 18th WMA General Assembly, Helsinki, Finland, June 1964—amended by the 64th WMA General Assembly, Fortaleza, Brazil, October 2013).

(3) Directive 2002/58/EC of the European Parliament and of the Council of 12 July 2002 concerning the processing of personal data and the protection of privacy in the electronic communications sector (directive on privacy and electronic communications)

(4) Regulation (EU) 2016/679 of the European Parliament and of the Council of 27 April 2016 on the protection of natural persons regarding the processing of personal data and on the free movement of such data and repealing Directive 95/46/EC (General Data Protection Regulation) (text with European Economci Area relevance).

(5) Regulation (EU) 2018/1725 of the European Parliament and of the Council of 23 October 2018 on the protection of natural persons regarding the processing of personal data by the union institutions, bodies, offices and agencies and on the free movement of such data and repealing Regulation (EC) No 45/2001 and Decision No 1247/2002/EC.

## Data analysis

Using the systematic literature review, a list of tentative attributes will be extracted from the thematic analysis based on gNBS themes and recurrent factors. The list will also include psychological factors (eg, depression, anxiety and emotion regulation) that will have to be addressed as contextual elements separately. As the literature extensively points out the psychosocial impact of gNBS, even in 2023 (Blom et al., 2023; Tobik et al., 2023), we decided to include the main psychological measures to ensure the possibility for subgroup analysis based on validated measures to better describe the sample but also be able to potentially describe differences in preferences within and between the populations.

As numerous potential attributes are expected to be highlighted during the literature review, the focus group will help clarify the most important elements. Independent coders will analyse the transcription of the focus group and provide content analysis. In order to refine the attribute list, the ranking task will be performed presenting 10–15 attributes to design the DCE. The data from the ranking task will be analysed to understand what attributes are most important to expecting parents. This information will be considered when deciding the final attributes for the DCE.

Participant characteristics that will be reported to describe the study samples include demographic characteristics (age, gender, pregnancy stage and income level, among others) but also attitudes and psychosocial factors such as health literacy and anxiety, for example (Hospital Anxiety and Depression Scale, available in both Italian and German; Annunziata et al., 2011; Hinz & Brähler, 2011).[41] [42] In collaboration with recruiting centres, we will consider the inclusion of personality scales, emotion regulation scales and/or coping strategy scales to provide a more precise psychological assessment.

Preferences and understanding of the survey instrument (comprehension questions) will be assessed by examining the relationship between psychological instruments of cognitive ability (eg, health literacy and numeracy) and survey responses. This includes comparing responses to warm up questions, survey internal validity checks and attitudinal questions about reported complexity of tasks across various levels of each psychological instrument.

To determine preferences of study participants regarding gNBS, attributes, level estimates and the conditional relative importance of attributes will be reported in the DCE using random parameters logit (RPL) modelling and latent class analyses (LCA). The heterogeneity of preferences and the impact of participant characteristics (eg, demographics and psychological validated scales) will be investigated by applying appropriate statistical models including LCA for the DCE and/or subgroup analyses (RPL). All results described above will be compared between the German and Italian study samples to determine whether there are cross-country differences. The results of the quantitative part of the preference study are planned to be published in a scientific journal (open access) and to be shared in all S4C project communication networks to reach out to different audiences as part of the general communication plan of the project.

## Author affiliations

[1]Center for Research and Bioethics, Uppsala Universitet, Uppsala, Sweden
[2]Research Unit of Neuromuscular and Neurodegenerative Disease, Ospedale Pediatrico Bambino Gesù IRCCS, Roma, Lazio, Italy
[3]Takeda Pharmaceuticals International AG, Opfikon, Zürich, Switzerland
[4]Medical Genetics, University of Siena, Siena, Italy
[5]Telethon Institute of Genetics and Medicine, Napoli, Campania, Italy
[6]Pfizer Inc, New York, New York, USA
[7]Medical Genetics Unit, Department of Medical Sciences, University of Ferrara, Ferrara, Italy
[8]Institute for Medical Humanities, First Faculty of Medicine, Charles University, Prague, Czech Republic
[9]EURORDIS, Paris, Ile-de-France, France
[10]Department of Neuropediatrics and Muscle Disorders, Medical Center, University of Freiburg, Faculty of Medicine, Freiburg, Germany
[11]Erlangen University Hospital, Erlangen, Bayern, Germany
[12]Bulgarian Association for Personalised Medicine, Sofia, Bulgaria
[13]Sanofi-Aventis SA, Diegem, Belgium
[14]Department of Neuropediatrics and Muscle Disorders, Faculty of Medicine, Freiburg, Germany
[15]Erasmus Universiteit Rotterdam, Rotterdam, Netherlands

**Contributors** SM, JC and JV drafted the protocol and closely ensured the adaptation of the final version to all authors's comments. SM, JC, JV, AG, BH and MH contributed to the design and implementation of the protocol. Planning of the work, conduct and reporting were ensured by all authors and included clinicians for the process of ethical application and procedure, but also to design the protocol in adequacy with clinical requirements. Ethical procedure was led by SM, MA, AE, ESB, LPB, AF, FF, VF, JK, FK, SO, AN, RR, CS, SS and NG. EG coordinated the patients and patient's advocacy participation for both ethical procedure and protocol drafting. SM, GK and JA drafted the first version of the project and included all relevant ethical consideration and introduction information. FK, RR, JK, SS, CS, LPB, VF, EG, JA, GK, FF, AF, NG, ESB, AE, SO, AN, SM, JC, JV, AG, BH and MH drafted the precise clinical information used in the draft, checked for relevance of the vocabulary and reviewed the overall structure of the article.

**Funding** This work was supported by S4C. This project has received funding from the IMI2 JU under grant agreement no. 101034427. The JU receives support from the European Union's Horizon 2020 research and innovation

programme and European Federation of Pharmaceutical Industries and Association.

**Competing interests** None declared.

**Patient and public involvement** Patients and/or the public were involved in the design, or conduct, or reporting, or dissemination plans of this research. Refer to the Methods section for further details.

**Patient consent for publication** Not applicable.

**Provenance and peer review** Not commissioned; externally peer reviewed.

**Data availability statement** No data are available.

**ORCID iDs**
Sylvia MARTIN http://orcid.org/0000-0002-3706-7669
Janbernd Kirschner http://orcid.org/0000-0003-1618-7386

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
