## [Reviewer comments · BMJ Open]

ARTICLE DETAILS

TITLE (PROVISIONAL)	Patient Preferences in Genetic Newborn Screening for Rare diseases: study protocol
AUTHORS	MARTIN, Sylvia; Angolini, Emanuele; Audi, Jennifer; Bertini, Dr. Enrico; Bruno, Lucia Pia; Coulter, Joshua; Ferlini, Alessandra; Fortunato, Fernanda; Frankova, Vera; Garnier, Nicolas; Grauman, Åsa; Gross, Edith; Hauber, Brett; Hansson, Mats; Kirschner, Janbernd; Knieling, Ferdinand; Kyosovksa, Gergana; Ottombrino, Silvia; Novelli, Antonio; Raming, Roman; Sansen, Stefaan; Saier, Christina; Veldwijk, Jorien

VERSION 1 – REVIEW

REVIEWER	Cowley, Lorraine Newcastle Upon Tyne Hospitals NHS Foundation Trust, Northern Genetics Service
REVIEW RETURNED	27-Nov-2023

GENERAL COMMENTS	This is an interesting and important collaborative project. I appreciate the challenges and benefits of writing with multiple international authors. The paper does have several typos throughout and would benefit from a proof read from a first language English speaker. I'm not sure of the abstract word count restrictions but it would be helpful to see the objectives in the abstract. I think using the term 'disability burden' (p5. Line 46) may be read negatively by people with disability. This may be a limitation of my knowledge about the method but my main concerns are around the Discrete Choice Experiment (DCE) methodology. In my opinion it would be difficult to distil the complexities and nuances of new-born genetic screening into bald statements of preference. The choice experiment is very limited because it is difficult to encapsulate all of the potential to act or not act on the outcomes of screening. How, for example, would reporting back incidental findings be represented to participants? Does the method corral respondents into making choices that do not exactly represent their opinions? How would the limitations of the method impact the findings? Whilst I accept that no research method can represent all participants' views, it would have been helpful to hear the authors acknowledging some of these issues. The authors have highlighted in the study limitations section that the choices in the methods are hypothetical and it would be good to see more discussion about this in the methods. The first introduction of the qualitative elements of the project pages 8-9 do not mention recruitment of healthy expecting parents. Although they are included later, my first impression was that they were excluded.
--

	The authors refer to the ‘General Population Sample’ (are these the healthy expecting parents?) being given a blocked design to permit completion of a subset of the choices. I would like to hear the authors’ views on whether this would create bias in the data. My understanding is that it would and I’d be interested to know how the authors would acknowledge and try to mitigate that bias. My knowledge of statistics is not honed enough to comment further and I have therefore requested a statistics review. The final attributes of the DCE methodology considered important to the anticipated target population are decided by the PLWRDs and Clinicians but healthy expecting parents (a significant proportion of the anticipated target population) are not represented in this decision making. I would like to hear the authors’ views on whether this creates bias. In the section concerning participant characteristics, the authors are considering incorporating collection of psychological assessment data. This is not flagged in the study objectives and no clear rationale is given about what additional value this would give the study or how it would potentially impact the findings. It would be helpful to have the rationale stated. The paper would benefit from minor revisions addressing these points. I look forward to hearing more about this project.
--	---

REVIEWER	Long, Janet Australian Institute of Health Innovation, Australian Institute of Health Innovation
REVIEW RETURNED	05-Dec-2023

GENERAL COMMENTS	This study is part of a larger “Screen4Care” project. The protocol describes the study as exploring attitudes and preferences towards neonatal blood spot screen using genomic technologies and AI to identify children that may have a rare disease. The “diagnostic odyssey” for people with a rare disease is well documented and provides a compelling rationale for this screening of neonates. Although data on preferences of the general community have been collected, this study will take a more granular look. They will conduct focus groups and interviews with expecting parents from two countries (Italy and Germany) divided by whether they are seeking testing or not. Interviews will also be conducted with medical geneticists. They will use DCE which is a sound methodology to compare different preferences and modes of delivery. Items will be developed based on findings from a systematic review. The rationale behind the study design is good. They have drawn on previous studies from the pre-natal testing field but recognise that neonatal testing has some similar but also some different priorities. Target sample sizes for the different groups is sound and the methodology is largely rigorous (see a couple of queries below). This is a huge amount of work (I hope they have a large enough team to make the data collection feasible) and I wish the researchers every success. A few comments PLWRD is a truly horrible acronym. I personally would write it out. “Person living with” is used as a respectful way of referring to the cohort, instead of e.g., “rare disease sufferers”. This respect is lost when you use the acronym. On Mss Page 8, line 37 you refer to an “external reviewer from Lucid”. What does this mean? Will need to explain to an international audience. Is a single reviewer doing the data
---

	extraction rather than the team? Can you justify this as rigorous practice? Perhaps it is just the way it is written? There is confusing and imprecise description of those taking part. It keeps changing. The aim of the study is given in several places (including the abstract) and needs to be standardised. You refer to two groups: “healthy” expecting parents in some places and “those seeking genetic testing” in others. It was not clear if the two groups of expectant parents were those opting in versus opting out of the testing or those with a family history of genetic condition/s versus those without a history. On page 10 – you talk about three groups of people taking part – here including the medical geneticists. Please standardise your description of the groups taking part across the abstract, introduction and methods. Also check the language around the different research activities – at one point I thought you were collecting the DCE data via the interviews. Needs to be clear and consistent. A graphic showing the different participants taking part and the data collection activities they will be participating in would be very useful indeed to clarify the overall design. Recruitment. The upper age limit of 70 (presumably for fathers and health professionals) provides a very broad range of ages. Will you be using age as a factor in your analysis. (I might have missed that). On page 12 the authors discuss contingency plan if their Ethics applications are not approved in one of the countries. Yet Ethics approval is stated in the abstract. I suggest you move this paragraph to page 15 where the staged Ethics approval is first explained. That puts these points into context. Typos and unclear wording: Mss P.5, Line 11 Words inverted Mss P.6, Line 18 It is a convention in academic writing to not have single sentence paragraphs. There are several places throughout the manuscript where this happens. Join these sentences to the end of the previous paragraph or start of the next paragraph. If that is not appropriate, add another sentence. Line 38 “...and also the early diagnoses possibilities” would be clearer stated as “...and also the possibility of earlier diagnosis.” Page 6, Line 43. Not clear, in fact this sentence is misleading. Please review the paper and rewrite. It was DCE – hypothetical factors that would or would not cause stress to the patients. Midwives wouldn’t know if something was a false positive so could not “choose to disclose”. Looks like you are saying false-positives are the same as findings that are not clinically relevant. Of course, these are very different. Page 7, Line 3 Last phrase is confusing. “A recent international research protocol (6 countries) addressed the preferences of women toward prenatal testing results [23] which has the same stepwise structure as our protocol in a connected field (prenatal testing).” Maybe “but in a connected field.”
--	--

REVIEWER	Mendes, Álvaro i3S, IBMC - University of Porto
REVIEW RETURNED	12-Dec-2023

GENERAL COMMENTS	Thank you for the opportunity to review this study protocol. This study holds significance as it aims to describe and measure the perspectives of key stakeholders regarding genetic newborn screening (gNBS) for rare diseases. The clear strengths of the proposed study include the ability to compare data from the general population in two different countries (Germany and Italy).
--

	Commendably, the authors have adopted a mixed-methods and multi-disciplinary approach to evaluation, and the involvement of patient organization representatives is a key strength. However, I have some reservations, primarily related to the organization of the study protocol. Firstly, throughout the document—Abstract, Introduction (page 6), Objectives (page 9)—the aims of the study are described in a somewhat confusing manner. It would be beneficial to present the aims coherently, perhaps through a subheading in the Introduction outlining the focus of the current study, and then another subheading in the Methods section titled "Research Aims and Objectives." Secondly, the authors need to clearly define what is understood by gNBS. Additionally, consistency is needed in the use of terminology—newborn screening or newborn testing?—throughout the manuscript, aligning with the title. Thirdly, the rationale for the study could be framed more clearly. While it is mentioned that there are ethical and social implications (pages 5 and 6), these aspects are not discussed in detail. I also suggest including more background information on how this study can impact expecting parents, parents seeking genetic testing, healthcare professionals, and the healthcare system more broadly. Crucially, there is no mention to existing guidelines on gNBS and how they are articulated with the study protocol. Additionally, enhancing the clarity of the study protocol could involve identifying and describing each study individually, akin to a "Detailed study plan," specifying the study design, data analysis, recruitment procedures, and sampling in more detail. This should be preceded by a contextualization of the overall research approach and conceptual framework. I have some additional questions: Was there any patient and public involvement in the drafting of the funding application for this project? Given that the findings from the studies will presumably be analyzed separately, do the authors plan to integrate the data at the end of the overall study to draw overarching conclusions, conducting a synthesis and interpretation study? Do the authors have plans regarding the dissemination of the study results (e.g., through traditional academic forums such as peer-reviewed publications, reports, study updates to collaborating patient organizations)? A discussion of the potential merits of this study, in terms of the methodology to be applied and the expected outcomes, is missing from the study protocol. I appreciate your attention to these suggestions and questions.
--	--

VERSION 1 – AUTHOR RESPONSE

Reviewer: 1

Dr. Lorraine Cowley, Newcastle Upon Tyne Hospitals NHS Foundation Trust

Comments to the Author:

This is an interesting and important collaborative project. I appreciate the challenges and benefits of writing with multiple international authors. The paper does have several typos throughout and would benefit from a proof read from a first language English speaker.

In order to ensure final proof-read, one native speaker will be in charge of final typos-proof reading (JC).

I'm not sure of the abstract word count restrictions but it would be helpful to see the objectives in the abstract.

We will be pleased to add a short sentence presenting the objectives (page 10 line 5 to 14) (page 8 line 24 to page 9 line 3), and counterbalanced the word count by withdrawing part of another sentence (page 3 line 6-7).

I think using the term 'disability burden' (p5. Line 46) may be read negatively by people with disability.

The term "burden" is used 3 times in draft, we suggest to replace it by different synonyms that will still represent the heaviness of the term but with more general vocabulary.

Page 6 line 12 (page 5 line 13) : difficulty

Page 6 line 25 (page 5 line 25): functional limitation

This may be a limitation of my knowledge about the method but my main concerns are around the Discrete Choice Experiment (DCE) methodology. In my opinion it would be difficult to distil the complexities and nuances of new-born genetic screening into bald statements of preference. The choice experiment is very limited because it is difficult to encapsulate all of the potential to act or not act on the outcomes of screening. How, for example, would reporting back incidental findings be represented to participants? Does the method corral respondents into making choices that do not exactly represent their opinions? How would the limitations of the method impact the findings? Whilst I accept that no research method can represent all participants' views, it would have been helpful to hear the authors acknowledging some of these issues.

The authors have highlighted in the study limitations section that the choices in the methods are hypothetical and it would be good to see more discussion about this in the methods.

We understand the doubts of the reviewer and have added text in two parts of the manuscript to provide more clarity. Where the qualitative phase will provide us with all arguments for and against participation in gNBS, the DCE will only determine the importance of the most important concepts of choice. While this leaves out several other concepts, this will not largely impact the study outcomes as has been shown in previous studies investigating the external validity and predictive power of this method.

In the listed limitations we have included (Page 4 line 6 to 9) (page 4 line 4 to 6):

"Only the most important characteristics of gNBS can be incorporated into the DCE, therefore considerations of less important characteristics can only be interpreted based on the outcomes of the qualitative work".

In the description of the DCE method we have included (page 10 line 29 to page 11 line 3) (page 9 line 18 to 22):

"Since a DCE can only include a total of about 5-7 most important attributes, it will not provide a fully holistic overview of the importance of all characteristics of gNBS on the decision to participate in such a screening. However, this is also not the aim of the method, using only the most important attributes results, uptake behavior can be correctly predicted as the positive predictive value of DCEs has been shown to be over 0.90."

A reference will be added in the clean version :

de Bekker-Grob, E. W., Donkers, B., Bliemer, M. C. J., Veldwijk, J., & Swait, J. D. (2020). Can healthcare choice be predicted using stated preference data? *Soc Sci Med*, 246, 112736. <https://doi.org/10.1016/j.socscimed.2019.112736>

The first introduction of the qualitative elements of the project pages 8-9 do not mention recruitment of healthy expecting parents. Although they are included later, my first impression was that they were excluded.

Page 12 line 1 and 2: a clarification is now added “testing and consultation” and “healthy expecting parents” separately.” In the clean version, the overall document has been triple checked to make sure that all occurrences have been clarified.

The authors refer to the ‘General Population Sample’ (are these the healthy expecting parents?) being given a blocked design to permit completion of a subset of the choices. I would like to hear the authors’ views on whether this would create bias in the data. My understanding is that it would and I’d be interested to know how the authors would acknowledge and try to mitigate that bias. My knowledge of statistics is not honed enough to comment further and I have therefore requested a statistics review.

Blocking designs is typical in DCEs and necessary to limit the cognitive burden of the survey (Johnson et al., 2013). Given the number of attributes and levels included in the DCE, respondents would not be able to complete a full factorial design (i.e., over 1000 choice tasks). The general rule of thumb is that each respondent needs to complete a minimum of choice tasks representing the number of expected parameters in the final analytical model plus 1. Therefore, most DCEs have respondents answer 8-15 choice questions. To increase the efficiency of the data collection, often a multiplicative of the number of choice tasks necessary per respondents is generated and divided over blocks. When blocking a design, all respondents will still see all attributes and levels but only a subset of combinations. The blocks of questions each respondent receives will be randomized to mitigate any potential bias of the blocked design while maximizing statistical power of the full design in the total study population (Hensher et al., 2015). (Page 12 line 22 to 33) (page 11 line 10 to 19)

The clean version of the draft will include the references:

Johnson, F. R., Lancsar, E., Marshall, D., Kilambi, V., Mühlbacher, A., Regier, D. A., ... & Bridges, J. F. (2013). Constructing experimental designs for discrete-choice experiments: report of the ISPOR conjoint analysis experimental design good research practices task force. *Value in health*, 16(1), 3-13.

Hensher, D., Rose, J. M., & Greene, W. H. (2015). *Applied Choice Analysis: second edition*. Cambridge University Press.

The final attributes of the DCE methodology considered important to the anticipated target population are decided by the PLWRDs and Clinicians but healthy expecting parents (a significant proportion of the anticipated target population) are not represented in this decision making. I would like to hear the authors’ views on whether this creates bias.

The final attributes will also be informed by perspectives of healthy expecting parents and expecting parents seeking genetic counseling during the qualitative portion of the study. The final attributes will be selected based on the results of the qualitative interviews and input from PLWRDs and Clinicians. This was mentioned but not clear we have added clarity to the manuscript the following places: page 3 line 13, page 11 line 6, page 12 line 16.

In the section concerning participant characteristics, the authors are considering incorporating collection of psychological assessment data. This is not flagged in the study objectives and no clear

rationale is given about what additional value this would give the study or how it would potentially impact the findings. It would be helpful to have the rationale stated.

We added page 22 line 15 to 19 (page 20 line 26 to 30):

“As the literature extensively points at the psychosocial impact of gNBS even in 2023 (Blom et al., 2023; Tobik et al., 2023), we decided to include the main psychological measures to ensure subgroup analysis based on validated measures to better describe the sample but also be able to potentially describe differences in preferences within and between the populations.”

The references will be added to the clean version of the manuscript:

Blom M, Bredius RGM, Jansen ME, Weijman G, Kemper EA, Vermont CL, Hollink IHIM, Dik WA, van Montfrans JM, van Gijn ME, Henriët SS, van Aerde KJ, Koole W, Lankester AC, Dekkers EHBM, Schielen PCJI, de Vries MC, Henneman L, van der Burg M; SONNET-Study Group. Parents' Perspectives and Societal Acceptance of Implementation of Newborn Screening for SCID in the Netherlands. *J Clin Immunol.* 2021 Jan;41(1):99-108. doi: 10.1007/s10875-020-00886-4. Epub 2020 Oct 18. PMID: 33070266; PMCID: PMC7846522.

Tobik K, Orland KM, Zhang X, Garcia K, Peterson AL. Parental Attitudes and Ideas Regarding Newborn Screening for Familial Hypercholesterolemia. *Matern Child Health J.* 2023 Jun;27(6):978-983. doi: 10.1007/s10995-023-03640-5. Epub 2023 Mar 25. PMID: 36964843.

The paper would benefit from minor revisions addressing these points. I look forward to hearing more about this project.

We thank Reviewer 1 for their comments and notes and stay at their disposition.

Reviewer: 2

Dr. Janet Long, Australian Institute of Health Innovation

Comments to the Author:

bmjopen-2023-081835, entitled "Patient Preferences in Genetic Newborn Testing for Rare diseases: study protocol."

This study is part of a larger "Screen4Care" project. The protocol describes the study as exploring attitudes and preferences towards neonatal blood spot screen using genomic technologies and AI to identify children that may have a rare disease. The "diagnostic odyssey" for people with a rare disease is well documented and provides a compelling rationale for this screening of neonates.

Although data on preferences of the general community have been collected, this study will take a more granular look. They will conduct focus groups and interviews with expecting parents from two countries (Italy and Germany) divided by whether they are seeking testing or not. Interviews will also be conducted with medical geneticists.

They will use DCE which is a sound methodology to compare different preferences and modes of delivery. Items will be developed based on findings from a systematic review.

The rationale behind the study design is good. They have drawn on previous studies from the pre-natal testing field but recognise that neonatal testing has some similar but also some different priorities.

Target sample sizes for the different groups is sound and the methodology is largely rigorous (see a couple of queries below). This is a huge amount of work (I hope they have a large enough team to make the data collection feasible) and I wish the researchers every success.

A few comments

PLWRD is a truly horrible acronym. I personally would write it out. "Person living with" is used as a

respectful way of referring to the cohort, instead of e.g., “rare disease sufferers”. This respect is lost when you use the acronym.

As S4C as a strong commitment to involve patients, this acronym was recommended to be used by our Patient’s Advisory Board led by EURORDIS. This acronym is not solely used by the S4C project nor EURORDIS. This term used in the UN Resolution to Increase Visibility for the 300 Million Persons Living with a Rare Disease, published on December 2021 (<https://www.rarediseasesinternational.org/wp-content/uploads/2022/01/Final-UN-Text-UN-Resolution-on-Persons-Living-with-a-Rare-Disease-and-their-Families.pdf>). We added a footnote (Page 6) to explain this choice of acronym p 6 “¹ This acronym has been selected based on its usage in international literature “<https://www.rarediseasesinternational.org/wp-content/uploads/2022/01/Final-UN-Text-UN-Resolution-on-Persons-Living-with-a-Rare-Disease-and-their-Families.pdf>”

On Mss Page 8, line 37 you refer to an “external reviewer from Lucid”. What does this mean? Will need to explain to an international audience. Is a single reviewer doing the data extraction rather than the team? Can you justify this as rigorous practice? Perhaps it is just the way it is written?

The exact process will be described in the SLR to be published separately. We described more specifically the process, bearing in mind that we cannot describe the methods or the SLR in the protocol description:

Page 11 line 12 to 18 (page 10 line 8 to 12): “11 duos of experts from the S4C project). Following the initial selection stages of articles, an external firm specializing in various databases and JBI tools, LUCID Ltd, located in Marlow, England, conducted a scientific quality evaluation (appraisal scoring). Subsequently, only those articles that exhibited a quality level above the average in their respective category were selected for further analysis.”

There is confusing and imprecise description of those taking part. It keeps changing. The aim of the study is given in several places (including the abstract) and needs to be standardized. You refer to two groups: “healthy” expecting parents in some places and “those seeking genetic testing” in others. It was not clear if the two groups of expectant parents were those opting in versus opting out of the testing or those with a family history of genetic condition/s versus those without a history. On page 10 – you talk about three groups of people taking part – here including the medical geneticists. Please standardize your description of the groups taking part across the abstract, introduction and methods.

In order to clarify the overall description of the study and population, we drafted a figure 3 (see bellow). Indeed, we will add clarification about the 2 different groups of “healthy/general population” vs “seeking genetic counseling”. As the first groups participating in the focus group will be recruited via our participating clinical centers in connection with the other services of their hospitals and clinics (on site) whereas, for the DCE, the participants will be also recruited from our clinical centers for parents seeking genetic counseling and also via a survey company for the “general public” (healthy a priori). This specific group has been entitled “general public” as we will broaden the recruitment range most probably to the country level but they are also expected to be “healthy” ones.

Also check the language around the different research activities – at one point I thought you were collecting the DCE data via the interviews. Needs to be clear and consistent.

The final version will be reviewed to ensure consistency.

A graphic showing the different participants taking part and the data collection activities they will be participating in would be very useful indeed to clarify the overall design.

We apologize for the non-visibility of the Figure 1 (page 15 line 4) (page 15 line 15 and subsequent) as it was supposed to present the procedure. We provide now in text 2 figures and added also a root version as a separate file (See flowcharts attached and page 27 and subsequent in draft).

Recruitment. The upper age limit of 70 (presumably for fathers and health professionals) provides a very broad range of ages. Will you be using age as a factor in your analysis. (I might have missed that).

In the DCE, demographics will be measured: age, education, income etc. As age of the mother has been pointed out to influence attitudes, age will be included in the analyses. Research identifies age to be correlated to maternal knowledge and attitudes towards gNBS for RDs (see classical example of demographics data used in Van der Pal et al., 2022). The age range was also deemed necessary for ethical approvals procedures in Italy but was more open in Germany (above 18 years old), so we decided to use the more restrictive description in the protocol description.

van der Pal SM, Wins S, Klapwijk JE, van Dijk T, Kater-Kuipers A, van der Ploeg CPB, Jans SMPJ, Kemp S, Verschoof-Puite RK, van den Bosch LJM, Henneman L. Parents' views on accepting, declining, and expanding newborn bloodspot screening. *PLoS One*. 2022 Aug 18;17(8):e0272585. doi: 10.1371/journal.pone.0272585. PMID: 35980961; PMCID: PMC9387838.

On page 12 the authors discuss contingency plan if their Ethics applications are not approved in one of the countries. Yet Ethics approval is stated in the abstract. I suggest you move this paragraph to page 15 where the staged Ethics approval is first explained. That puts these points into context.

We rewrote the paragraph (page 15 line 32) (page 14 line 29 to 31) as we wanted to state that the ethical application could not be exactly duplicated in 2 separate countries as some domestic standards made us refine for example in Italy, the type of professional that could perform the interviews with patients at the clinical centers. "Methods have been adapted for example by conducting individual interviews for parents seeking genetic testing (more sensitive sample) only by clinical geneticist (in Italy)."

Typos and unclear wording:

Mss P.5, Line 11 Words inverted

Mss P.6, Line 18 It is a convention in academic writing to not have single sentence paragraphs. There are several places throughout the manuscript where this happens. Join these sentences to the end of the previous paragraph or start of the next paragraph. If that is not appropriate, add another sentence.

The overall document has been double checked for typos and single line paragraphs.

Line 38 "...and also the early diagnoses possibilities" would be clearer stated as "...and also the possibility of earlier diagnosis."

The change has been made (now page 8 line 12) (page 7 line 11).

Page 6, Line 43. Not clear, in fact this sentence is misleading. Please review the paper and rewrite. It was DCE – hypothetical factors that would or would not cause stress to the patients. Midwives wouldn't know if something was a false positive so could not "choose to disclose". Looks like you are saying false-positives are the same as findings that are not clinically relevant. Of course, these are very different.

Page 8 line 15 (page 7 line 13 to 16): we rephrased the sentence to ensure that the meaning is clear: “they answered that the potential for receiving a “false positive” result should not be disclosed (as it may not be helping decision making), and that the best period to provide information was late pregnancy to 3 days post-birth.”

Page 7, Line 3 Last phrase is confusing. “A recent international research protocol (6 countries) addressed the preferences of women toward prenatal testing results [23] which has the same stepwise structure as our protocol in a connected field (prenatal testing).” Maybe “but in a connected field.”

Change has been made page 9 line 5 (page 8 line 3).

Best wishes for your future research,
JL

Reviewer: 3

Dr. Álvaro Mendes, i3S, IBMC - University of Porto

Comments to the Author:

Thank you for the opportunity to review this study protocol. This study holds significance as it aims to describe and measure the perspectives of key stakeholders regarding genetic newborn screening (gNBS) for rare diseases. The clear strengths of the proposed study include the ability to compare data from the general population in two different countries (Germany and Italy). Commendably, the authors have adopted a mixed-methods and multi-disciplinary approach to evaluation, and the involvement of patient organization representatives is a key strength.

However, I have some reservations, primarily related to the organization of the study protocol. Firstly, throughout the document—Abstract, Introduction (page 6), Objectives (page 9)—the aims of the study are described in a somewhat confusing manner. It would be beneficial to present the aims coherently, perhaps through a subheading in the Introduction outlining the focus of the current study, and then another subheading in the Methods section titled “Research Aims and Objectives.”

As requested by Reviewer 1, the abstract will be amended with a mention of the objectives (bearing in mind the word count).

We regrouped all Aims and Objectives under the subheading “Research Aims and Objectives” page 9 line 21 (page 8 line 15).

Secondly, the authors need to clearly define what is understood by gNBS.

We added a clarification of terms coming from the definition of the Institute NHGR. 2023, 17th November [Available from: <https://www.genome.gov/genetics-glossary/Newborn-Genetic-Screening>. A footnote page 6 (page 5): “We will use a broad definition of genetic newborn screening as defined by Institute NHGR. 2023, 17th November [Available from: <https://www.genome.gov/genetics-glossary/Newborn-Genetic-Screening> : “*Newborn screening is a set of laboratory tests performed on newborn babies to detect a set of known genetic diseases. Typically, this testing is performed on a blood sample obtained from a heel prick when the baby is two or three days old. [...].*”

The reference will be added to the final list of references in the final draft approval stage.

Additionally, consistency is needed in the use of terminology—newborn screening or newborn testing?—throughout the manuscript, aligning with the title.

The focus of the preferences study will be on participation in genetic newborn screening, as a consequence of this screening, there will be a genetic test to confirm any potential diagnosis.

Therefore, we will use the term “genetic screening test” when referring to the recruitment.

We added a definition footnote page a6 to clarify the definition of newborn genetic screening that will also improve the overall understanding of the terms (see above).

Thirdly, the rationale for the study could be framed more clearly. While it is mentioned that there are ethical and social implications (pages 5 and 6), these aspects are not discussed in detail.

We added the below comment on the fact that ethical and social implication of gNBS are debated.

(Page 6 line 5 to 9) “Ethical and psycho-social issues related to newborn screening present considerable dilemmas regarding the effects of genetic data on both the infant and their family, as well as on broader societal aspects. These concerns play a critical role in shaping regulatory structures and health policies aimed at addressing these ethical obstacles (Reinsteing, 2015; Grob, 2019).”

Grob R. Qualitative Research on Expanded Prenatal and Newborn Screening: Robust but Marginalized. *Hastings Cent Rep.* 2019 May;49 Suppl 1(Suppl 1):S72-S81. doi: 10.1002/hast.1019. PMID: 31268576; PMCID: PMC8115092.

Reinstein E. Challenges of using next generation sequencing in newborn screening. *Genetics Research.* 2015;97:e21.

The reference in text and in the reference section will be amended accordingly in the latest version of the draft.

I also suggest including more background information on how this study can impact expecting parents, parents seeking genetic testing, healthcare professionals, and the healthcare system more broadly.

In the introduction section we suggest (page 7 line 18 to 23): “Moreover, this study will allow us to assess the perspective of gNBS in Europe among expecting parents or individuals from general public who might face such a decision as health care systems are on the verge of implementing these programs. In fact, most of the European countries that implemented neonatal screening in the 60s/ 70s are now implementing an extended panel, envisioning the potential to screen for 40 to 50 conditions to be tested with a single blood spot because of the technological improvements (molecular technologies) [22].” And develops on how the research is important for the field.

We added the following in the introduction section:

Page 7 line 23 to 30: “Protocols for genetic newborn screening research offer innovations for prompt and efficient treatment, as well as reducing obstacles related to worry in parents or over diagnosing (Gray et al., 2008). These studies serve as a catalyst for ongoing developments in screening's sensitivity and specificity. The goal of the screening process optimization efforts is to provide assurance that true positive cases will be accurately diagnosed and to prevent families from being upset by an unexpected false positive result or from experiencing severe anxiety as a result of results of uncertain significance (La Marca et al., 2023)”.

Crucially, there is no mention to existing guidelines on gNBS and how they are articulated with the study protocol.

We added a references to existing references from National Human Genome Institute that emphasize the importance of ethical issues and articulated with the present protocol

Page 7 line 7 (page 6 line 9 to 17) : “Moreover, if we refer to the National Human Genome Research Institute *Newborn Screening Fact Sheet*, “with the decreasing cost of genome sequencing, there is potential for its clinical application in newborn screening. This could supplement or replace traditional panels of tests, providing more comprehensive health information. However, several questions remain about the practicality, ethics, and long-term implications of incorporating genome sequencing into routine newborn screening” (<https://www.genome.gov/about-genomics/fact-sheets/Newborn-Screening-Fact-Sheet>, consulted on January 2d 2024). Therefore, clarifying HCPs, patients and general public perspectives on gNBS implication will improve its safe and fair development.”

References will be amended accordingly in the latest version of the draft.

Additionally, enhancing the clarity of the study protocol could involve identifying and describing each study individually, akin to a "Detailed study plan," specifying the study design, data analysis, recruitment procedures, and sampling in more detail.

Figure 3 will be added.

This should be preceded by a contextualization of the overall research approach and conceptual framework.

We added a Figure to clarify the recruitment procedures and different phases of the protocol (in situ versus online, qualitative vs quantitative).

I have some additional questions:

Was there any patient and public involvement in the drafting of the funding application for this project?

Patient advisory board, as representative of patients lead by Eurordis, participated in the design and drafting and were included in the review of all ethical application and information material design. Moreover close collaboration for the subsequent part of the study will be ensure as patient representatives will be included at every step of the protocol.

Given that the findings from the studies will presumably be analyzed separately, do the authors plan to integrate the data at the end of the overall study to draw overarching conclusions, conducting a synthesis and interpretation study?

The analysis is planned to be separated in at least 2 different articles. A first article for qualitative output (based on focus group data) and second article will look at the DCE results. However, the content of the DCE will be fully informed by the qualitative work (which will also be describe in the respective paper).

Moreover, the preliminary work of the literature review has been performed at the systematic level and will be published alongside the analysis papers.

Do the authors have plans regarding the dissemination of the study results (e.g., through traditional academic forums such as peer-reviewed publications, reports, study updates to collaborating patient organizations)?

The dissemination plan will be both academic (scientific publication) and for patient organization in conferences and seminars but also potentially shared via our S4C dissemination network (Eurice).

A discussion of the potential merits of this study, in terms of the methodology to be applied and the expected outcomes, is missing from the study protocol.

These elements were part of the Strengths and Limitation section but they are not fitted for the structure of the paper as the Editor noted “The novelty, aims, results or expected impact of the study should not be summarized here.”

We will clarify this in the introduction as follows: This study will inform both clinical research and patient advocates stakeholders contributing to gNBS for rare diseases field.

I appreciate your attention to these suggestions and questions.

We thank Reviewer 3 for the comments and suggestions.

VERSION 2 – REVIEW

REVIEWER	Cowley, Lorraine Newcastle Upon Tyne Hospitals NHS Foundation Trust, Northern Genetics Service
REVIEW RETURNED	29-Jan-2024

GENERAL COMMENTS	Thank you for addressing all of the points so thoroughly. I am looking forward to the findings and outcomes of this interesting study. Best wishes and good luck in your collaborations.
--

REVIEWER	Long, Janet Australian Institute of Health Innovation, Australian Institute of Health Innovation
REVIEW RETURNED	13-Feb-2024

GENERAL COMMENTS	Many thanks to the authors for addressing all my comments. The methods are much clearer now aided in particular by the excellent graphics. Thanks for the explanation of the PLWRD acronym too. Best wishes, JL
--

REVIEWER	Mendes, Álvaro i3S, IBMC - University of Porto
REVIEW RETURNED	20-Feb-2024

GENERAL COMMENTS	Thank you for addressing all of my comments and questions.
--